# Mycotoxins in Flanders’ Fields: Occurrence and Correlations with *Fusarium* Species in Whole-Plant Harvested Maize

**DOI:** 10.3390/microorganisms7110571

**Published:** 2019-11-18

**Authors:** Jonas Vandicke, Katrien De Visschere, Siska Croubels, Sarah De Saeger, Kris Audenaert, Geert Haesaert

**Affiliations:** 1Department of Plants and Crops, Faculty of Bioscience Engineering, Ghent University, Valentin Vaerwyckweg 1, 9000 Ghent, Belgium; Kris.Audenaert@UGent.be; 2Biosciences and Food Sciences Department, Faculty Science and Technology, University College Ghent, Research Station HoGent-UGent, Diepestraat 1, 9820 Bottelare, Belgium; katrien.devisschere@hogent.be; 3Department of Pharmacology, Toxicology and Biochemistry, Faculty of Veterinary Medicine, Ghent University, Salisburylaan 133, 9820 Merelbeke, Belgium; Siska.Croubels@ugent.be; 4Department of Bio-analysis, Faculty of Pharmaceutical Sciences, Ghent University, Ottergemsesteenweg 460, 9000 Ghent, Belgium; sarah.desaeger@ugent.be

**Keywords:** Maize, mycotoxins, *Fusarium*, monitoring, forage, silage, maize ear rot, nivalenol, fumonisins

## Abstract

Mycotoxins are well-known contaminants of several food- and feedstuffs, including silage maize for dairy cattle. Climate change and year-to-year variations in climatic conditions may cause a shift in the fungal populations infecting maize, and therefore alter the mycotoxin load. In this research, 257 maize samples were taken from fields across Flanders, Belgium, over the course of three years (2016–2018) and analyzed for 22 different mycotoxins using a multi-mycotoxin liquid chromatography-tandem mass spectrometry (LC-MS/MS) method. DNA of *Fusarium graminearum*, *F. culmorum* and *F. verticillioides* was quantified using the quantitative polymerase chain reaction (qPCR). Multi-mycotoxin contamination occurred frequently, with 47% of samples containing five or more mycotoxins. Nivalenol (NIV) was the most prevalent mycotoxin, being present in 99% of the samples, followed by deoxynivalenol (DON) in 86% and zearalenone (ZEN) in 50% of the samples. Fumonisins (FUMs) were found in only 2% of the samples in the wet, cold year of 2016, but in 61% in the extremely hot and dry year of 2018. Positive correlations were found between DON and NIV and between *F. graminearum* and *F. culmorum*, among others. FUM concentrations were not correlated with any other mycotoxin, nor with any *Fusarium* sp., except *F. verticillioides*. These results show that changing weather conditions can influence fungal populations and the corresponding mycotoxin contamination of maize significantly, and that multi-mycotoxin contamination increases the risk of mycotoxicosis in dairy cattle.

## 1. Introduction

Ensiling forage crops is a common way of ensuring a continuous and stable supply of feed throughout the year in dairy husbandry. These silages, mostly grass or maize [1], represent 50–80% of the diet of dairy cows during the winter [2]. Especially in North-Western Europe, fodder maize cultivation for on-farm use is an essential part of dairy husbandry [3]. In the region of Flanders, Belgium, more than 127,000 ha of silage maize was grown in 2018, making it the second most grown crop behind pasture [4].

Maize silages can be contaminated with mycotoxins, secondary metabolites produced by a variety of moldy fungi. Mycotoxins can cause several acute and chronic toxic effects to humans and animals when ingested.

In general, ruminants are less sensitive to mycotoxins than monogastrics due to the ability of the ruminal flora to degrade several mycotoxins to less toxic substances [5]. However, not all mycotoxins are degraded in this way. Some can be converted to molecules with a higher toxicity level (e.g., zearalenone (ZEN) to α-zearalenol (α-ZEL)), while others are not even converted at all (e.g., fumonisins (FUMs)) [6,7,8,9]. Moreover, if both the rumen microflora and the pH are not stable (e.g., in calves, high-yielding cows or animals in the transition period), mycotoxin metabolism is reduced [10]. Therefore, dairy cattle are susceptible to mycotoxic effects as well, including gastroenteritis, reduced feed intake and reduced fertility [10,11,12], leading to economic losses [13].

Mycotoxins can be produced in the field (preharvest) as well as in the silage (postharvest). In preharvest field conditions within temperate regions, mycotoxins are mainly produced by *Fusarium* spp., causing maize stem and ear rot [1,14,15,16]. Two main types of maize ear rot can be distinguished: red ear rot (or *Gibberella* ear rot), primarily caused by *F. graminearum*, *F. culmorum* and *F. poae*, and pink ear rot (or *Fusarium* ear rot), primarily caused by *F. verticillioides, F. proliferatum* and *F. subglutinans* [17,18,19,20]. The distribution and prevalence of these *Fusarium* spp. is dependent upon geography and climate. In Europe, the most isolated *Fusarium* species are *F. graminearum* and *F. culmorum*, dominantly in the North, and *F. verticillioides*, mostly found in the South [19,21,22,23]. However, maize ear rot is always caused by a *Fusarium* complex, rather than by a single species [16,19,24]. Different *Fusarium* spp. interact with each other, leading to possible synergistic effects for infection, although reports have been contradictory [25,26,27,28,29]. Different pathways can be used by *Fusarium* spp. to infect maize plants, and while some *Fusarium* spp. prefer a primary infection via the silks (e.g., *F. graminearum*), others use a systemic transmission from root to kernel, or co-occur as a secondary infection when insects damage the kernels (e.g., *F. verticillioides*) [30,31,32,33,34]. This makes the prevention and control of *Fusarium* spp. in maize very difficult and complex.

During infection, *Fusarium* spp. can produce a variety of mycotoxins. Some of the most well-known *Fusarium* mycotoxins include deoxynivalenol (DON), causing reduced feed intake and diarrhea; zearalenone (ZEN), causing fertility problems; and the fumonisins (FUMs), causing liver and kidney injuries. Other important *Fusarium* mycotoxins include nivalenol (NIV), T-2 toxin (T2), diacetoxyscirpenol (DAS) and enniatins (ENN), among others [11,12,35,36]. Mycotoxins produced by other fungal species, such as aflatoxins produced by *Aspergillus* spp., are rarely found in temperate climates [37,38,39,40]. However, climate change may influence the geographical spread of mycotoxin-producing fungi in Europe, causing more tropical fungi, such as *Aspergillus flavus* and *Fusarium verticillioides*, to migrate northward [23,41,42,43,44,45,46].

*Fusarium* spp. cannot survive postharvest silage conditions if the silage is firmly pressed and sealed hermetically, but *Fusarium* mycotoxins are stable molecules that may remain unchanged during the silage process [24,47,48,49,50]. If a silage is not pressed and sealed correctly and oxygen remains present, *Fusarium* spores may germinate and colonize the maize silage and produce additional mycotoxins [51,52,53]. Furthermore, other fungal species such as *Penicillium* spp. and *Aspergillus* spp. are well adapted to the silage conditions and may produce additional mycotoxins [47,54,55,56,57].

This cocktail of mycotoxins, coming from different fungal species, has led to the observation that almost every maize or maize silage sample is contaminated with at least one mycotoxin, and often multiple. Numerous surveys have been conducted in many regions in the world [22,24,37,38,39,40,48,54,58,59,60,61,62]. However, most of these surveys were conducted on samples of maize ears, rather than on the entire plant. Some reports state that the ear can be a representative sample for the entire plant [38], although the fungal species composition and mycotoxin concentrations can differ [16]. Furthermore, most surveys focused on a selection of mycotoxins, rather than screening the entire mycotoxin load. Severe multi-mycotoxin contaminations could hence be overlooked.

The European Union (EU) has set a maximum level for aflatoxin B1 (AFB1) [63] and guidance values for DON, ZEN, ochratoxin A (OTA), fumonisin B1 (FB1) and B2 (FB2), T-2 toxin (T2) and HT-2 toxin (HT2) in several food- and feedstuffs, including maize [64,65]. No recommendations have been formed on lesser researched mycotoxins like NIV, modified mycotoxins like 3- or 15-acetyldeoxynivalenol (3-ADON and 15-ADON), or emerging mycotoxins like enniatins (ENN) [66,67].

Neither do these guidance values take into account any possible synergistic effects of multi-mycotoxin contamination [40,62,68,69,70,71,72]. As a result, one cannot assess whether a particular feed sample is safe, based on these guidance values alone [2]. A better strategy to safeguard livestock health would be to avoid fungal infection and the production of mycotoxins in the first place.

The aim of this research was to investigate the natural mycotoxin load in harvested maize plants intended for silage in the Northwestern European region of Flanders over the course of three years, and link these concentrations to the presence of certain mycotoxigenic *Fusarium* species. A total of 257 samples were taken from harvested maize fields across Flanders during 2016–2018. Samples were analyzed for 22 different mycotoxins using a multi-mycotoxin liquid chromatography-tandem mass spectrometry (LC-MS/MS) method. Then, using a quantitative polymerase chain reaction (qPCR), the DNA of three of the most prevalent *Fusarium* spp. in Flanders, namely *F. graminearum, F. culmorum* and *F. verticillioides* [21,73,74], was quantified in the same maize samples. With these data, we were able to quantify the mycotoxin load of silage maize fields in practice, compare mycotoxin occurrence between different years and weather conditions, and identify correlations between these mycotoxins and the corresponding *Fusarium* spp.

## 2. Materials and Methods

### 2.1. Maize Sampling

A total of 106 dairy farmers across Flanders were contacted to participate in this study from 2016 till 2018. The selected maize fields were scattered throughout Flanders, grown on different soils and in different micro climates, and the number of selected fields was proportional to the intensity of maize production in that region. Data regarding daily temperature, rainfall, relative humidity and radiation for each growing season were obtained from 17 weather stations spread across Flanders (Figure 1). A few months before harvest, each farmer received a plastic bag, a label and a manual, in which the sampling technique was explained. Sampling was done by taking at least 10 samples per trailer of harvested maize. These samples were mixed and a subsample of ca. 1 kg was put into a plastic bag. The bag was then sealed airtight and stored in a freezer until it was collected by the researchers. After sample collection, a subsample of ca. 5 g was taken for quantitative polymerase chain reaction (qPCR) analysis and stored in a freezer at –20 °C until further analysis; the remaining sample was dried in an airstream of 65 °C for four days. The dried maize sample was then milled in a 0.5 mm sieve, and stored until further mycotoxin analysis.

### 2.2. Reagents and Chemicals for LC-MS/MS

Methanol (LC-MS grade), glacial acetic acid (LC-MS grade), and analytical grade acetonitrile were purchased from Biosolve B.V. (Valkenswaard, The Netherlands). Analytical grade acetic acid and ammonium acetate were obtained from Merck (Darmstadt, Germany), while analytical grade n-hexane and methanol were purchased from VWR International (Zaventem, Belgium). Water was purified using a Milli-Q Gradient System (Millipore, Brussels, Belgium).

Certified mycotoxin standard solutions in acetonitrile of OTA (10 µg/mL), aflatoxin mix (AFB1, AFB2, AFG1 and AFG2) (20 µg/mL), fumonisin mix (FB1 and FB2) (50 µg/mL), sterigmatocystin (STERIG) (50 µg/mL), DON (100 µg/mL), deepoxy-deoxynivalenol (DOM) (50 µg/mL), ZEN (100 µg/mL), NIV (100 µg/mL), neosolaniol (NEO) (100 µg/mL), T2 (100 µg/mL), 3-ADON (100 µg/mL), DAS (100 µg/mL), 15-ADON (100 µg/mL), and F-X (100 µg/mL) were obtained from Romer Labs (Tulln an der Donau, Austria). Alternariol (AOH), alternariol monomethylether (AME), zearalanone (ZAN) and enniatin B (ENN B) were purchased from Sigma-Aldrich (Bornem, Belgium), fumonisin B3 was obtained from Promec unit (Tygerberg, South Africa), and roquefortine C (ROQ-C) was purchased from Alexis Biochemicals (Enzo Life Sciences BVBA, Zandhoven, Belgium). Stock solutions were prepared in acetonitrile/water (50/50 *v*/*v*) for FB3 (1 mg/ml), methanol/dimethylformamide (60/40 *v*/*v*) for AOH and AME (1 mg/ml), and methanol for ROQ-C and ZAN (1 mg/ml). All stock solutions were stored at –20 °C, except for the stock solution of FB3 (4 °C). Working solutions were prepared by diluting the stock solutions in methanol, and were stored at –20 °C for three months. Three standard mixture working solutions were prepared in methanol. The first contained the mycotoxins AFB1, AFB2, AFG1 and AFG2 (2 ng/µL); OTA (5 ng/µL); ZEN and T2 (10 ng/µL); and DON, FB1 and FB2 (40 ng/µL). The second contained DAS (0.5 ng/µL); ROQ-C (1 ng/µL); 15-ADON (2.5 ng/µL); 3-ADON and STERIG (5 ng/µL); NEO and AOH (10 ng/µL); NIV, F-X and AME (20 ng/µL); and FB3 (25 ng/µL). The third contained ENN B (10 ng/µL). The standard mixtures were stored at –20 °C and used for a maximum of three months.

### 2.3. Sample Preparation for LC-MS/MS

Twenty-two mycotoxins were extracted from the samples according to the methodology described by Monbaliu et al. [75]. Five grams of dried maize sample was spiked with internal standards ZAN and DOM at a concentration of 200 and 250 µg/kg, resp. The spiked sample was kept in the dark for 15 min and extracted with 20 ml of extraction solvent (acetonitrile/water/acetic acid (79/20/1, *v*/*v*/*v*)), and then agitated on a vertical shaker for 1 h. After centrifuging for 15 min at 3300 g, the supernatant was passed through a preconditioned C18 solid phase extraction (SPE) column (Alltech, Lokeren, Belgium). The eluate was diluted to 25 ml with extraction solvent and defatted with 10 ml n-hexane. In order to recover all 22 mycotoxins, two different clean-up pathways were followed. In the first pathway, 10 ml of extract was diluted with 20 ml acetonitrile/acetic acid (99/1 *v*/*v*), passed through a Multisep^®^226, AflaZon+ multifunctional column from Romer Labs (Tulln, Austria) and washed with 5 ml acetonitrile/acetic acid (99/1 *v*/*v*). For the second pathway, 10 ml extract was filtered using a Whatman glass microfilter (VWR International, Zaventem, Belgium). Two milliliters of this filtered extract was combined with the MultiSep 226 eluate from the first pathway and evaporated to dryness. The residue was then redissolved into 150 µL of mobile phase (water/methanol/acetic acid (57.2/41.8/1, *v*/*v*/*v*)) and 5 mM ammonium acetate. Lastly, the solution was centrifuged for 5 min at 14,000 *×g* using ultra free-MC centrifuge filters (Millipore, Bedford, MA, USA).

### 2.4. Mycotoxin Analysis by LC-MS/MS

The samples were analyzed using a micromass Quattro Premier XE triple quadrupole mass spectrometer coupled with a Waters Acquity UPLC system (Waters, Milford, MA, USA). Data processing was done using the Masslynx^TM^ (4.1 version) and Quanlynx^®^ (4.1 version) software (Micromass, Manchester, UK). The analytical column used was a Symmetry C18, 5 µm, 2.1 × 150 mm, with a guard column of the same material (3.5 µm, 10 mm × 2.1 mm) (Waters, Zellik, Belgium) kept at room temperature. The injection volume was 10 µL. Capillary voltage was set at 3.2 kV with a source block temperature and desolvation temperature of 120 and 400 °C, resp. Liquid chromatography conditions and MS parameters were followed as described by Monbaliu et al. [75].

### 2.5. LC-MS/MS Quality Control

To compensate for matrix effects and losses during extraction and cleanup, DOM (a structural analogue of DON) and ZAN (a structural analogue of ZEN) were used as internal standards. For each mycotoxin, five blank samples were spiked at five concentration levels. A cutoff (CO) level was established for every mycotoxin. The CO levels were based on the current regulatory levels, if available [64,65,76]; else, the CO level was chosen arbitrarily. The decision limit CCα was defined as the concentration at the y-intercept plus 2.33 times the standard deviation (SD) of the within lab reproducibility (α = 1%). The apparent recovery was calculated by dividing the observed value from the calibration plot by the spiked level. Linearity was tested graphically using a scatter plot, and the linear regression model was evaluated using a lack-of-fit test.

LOD and LOQ were estimated for each separate mycotoxin using the blank samples spiked at five different concentrations, which provided a signal-to-noise (S/N) ratio of 3 and 10, resp., in accordance to the definitions set by the International Union of Pure and Applied Chemistry (IUPAC). The interday repeatability was calculated using the relative standard deviation (RSD) at the spiked concentration levels.

### 2.6. qPCR Analysis

A quantitative PCR (qPCR) assay was used to quantify the total *F. graminearum*, *F. verticillioides* and *F. culmorum* DNA content in the maize samples from 2017 and 2018. In 2016, no samples for qPCR were taken. These three species were selected based on the known fungal species composition in temperate climates and in Belgium in particular [21,35,73,74], and to cover most *Fusarium* producers of mycotoxins that were included in the LC-MS/MS analysis [15,35,77]. Each subsample (5 g) was crushed with liquid nitrogen using a pestle and mortar and approx. 150 mg (the exact amount was weighted) was transferred to a 1.5 ml Eppendorf tube for DNA extraction. DNA was extracted from harvested maize samples using a CTAB method modified for use with fungi [78]. The total amount of DNA was quantified with a Quantus fluorometer (Promega, Leiden, The Netherlands), and stored at –20 °C. Then qPCR analysis was performed. The qPCR mix consisted of 6.25 µL of GoTaq® qPCR Master Mix (Promega, Leiden, The Netherlands), the corresponding primers (0.625 µL primer, 5 µM), 2 µL of DNA, 0.208 µL CXR reference dye (Promega, Leiden, The Netherlands), and watered to 12 µL. The used primers were FgramB379 forward (CCATTCCCTGGGCGCT), FgramB411 reverse (CCTATTGACAGGTGGTTAGTGACTGG), FculC561 forward (CACCGTCATTGGTATGTTGTCACT), FculC614 reverse (CGGGAGCGTCTGATAGTCG), Fver356 forward (CGTTTCTGCCCTCTCCCA), and Fver412 reverse (TGCTTGACACGTGACGATGA) [79]. The qPCR analysis was performed using a CFX96 system (Bio-Rad, Temse, Belgium), including the following thermal settings: 95 °C for 3 min; 40 cycles of 95 °C for 10 s, and 60 °C for 30 s, followed by dissociation curve analysis at 65 to 95 °C.

### 2.7. Statistical Analysis

The Pearson correlation coefficient was used to detect relations between different mycotoxins, between mycotoxins and fungal DNA, and between different *Fusarium* spp. at a significance level of *p* = 0.05. For calculation of the correlation coefficients, four outliers were discarded in the *F. verticillioides* DNA data and one in the *F. graminearum* DNA data. All statistical analyses were conducted using the R software package (R Core Team, Vienna, Austria) version 3.4.3 [80].

## 3. Results

### 3.1. Mycotoxin Levels in Harvested Maize Samples in 2016–2018

Incidence, mean, median and maximum concentrations, and the numbers of samples exceeding the European regulations can be found in Table 1; Complete results per sample can be found in Appendix A. NIV was the most prevalent mycotoxin, being present in 99.2% of all samples between 2016 and 2018. DON was present in all samples in 2017, but only in 64.7% of the samples in 2018. Over the three years, DON and its derivates 3-ADON and 15-ADON (described together as DON+) were the second most prevalent mycotoxins. ZEN’s highest incidence was in 2016, with 64.8% of the samples contaminated, while only 40.7% and 42.4% of the samples were contaminated in 2017 and 2018, resp. FB1, FB2 and FB3 incidence rose considerably from 2016 till 2018, with a total fumonisin incidence (described as FUM) of only 2.5% in 2016, to 19.8% in 2017 and 61.2% in 2018. AOH, AME, DAS, FX, T2, STERIG and ROQ-C were detected sporadically and never reached incidences higher than 11.0%. NEO, AFB1, AFB2, AFG1, AFG2 and OTA were never detected.

Mean concentration of NIV rose from 650.7 µg/kg in 2016, to 719.0 µg/kg in 2017 and 881.9 µg/kg in 2018. The highest mean concentration of DON was found in 2017 (557.5 µg/kg), while the lowest concentration was found in 2018 (186.5 µg/kg). Concentrations for NIV and DON went as high as 6776.3 µg/kg and 5322.5 µg/kg, resp. These concentrations were detected in the same sample from a maize field in 2017. This sample contained the highest total mycotoxin load of all years, with a total mycotoxin concentration of 13,747.6 µg/kg. Mean concentrations of fumonisin (FUM) rose simultaneously with its incidence, from 1.3 µg/kg in 2016 to 327.0 µg/kg in 2018. The average total mycotoxin load in a maize sample from 2016 till 2018 was 1692.0 µg/kg. Over the three years, 2.3% and 7.8% of the maize samples exceeded the EU guidance values for DON and ZEN, resp. No samples exceeded the guidance values for FB1, FB2, FB3 or T2, nor the maximum level for AFB1.

A vast majority of the maize samples was contaminated with more than one mycotoxin. Only one out of the 257 samples analyzed over the course of three years contained no mycotoxins, while 46.7% of all samples contained five or more mycotoxins. The median load was four mycotoxins per sample. When comparing the multi-mycotoxin contamination of each year (Figure 2), it is clear that the most diversely contaminated maize samples were found in 2017 and 2018. In 2018, two samples even contained 10 different mycotoxins. In 2016, the maximum number of detected mycotoxins in one sample was seven. However, 62.6% of the samples in 2016 contained five or more different mycotoxins, leading to the highest median mycotoxin load per sample (five mycotoxins per sample, compared to four mycotoxins per sample for 2017 and 2018, resp.).

### 3.2. Correlations between Different Mycotoxins

A heat map with correlations between different mycotoxins for 2016–2018 is shown in Figure 3. NIV was significantly correlated with DON (r = 0.38, *p* < 0.001) and its derivates 3-ADON (r = 0.22, *p* < 0.001) and 15-ADON (r = 0.28, *p* < 0.001). NIV was also significantly correlated with ZEN and ENN B, although the correlations were rather weak (r = 0.21, *p* < 0.001 and r = 0.12, *p* = 0.0496, resp.). Other correlations were non-existent or not significant. When splitting the data per year, similar results were obtained, however some differences occurred (Figure A1, Figure A2 and Figure A3). For instance, the correlation between NIV and DON was the strongest in 2017 (r = 0.65, *p* < 0.001), but was not significant in 2016 and 2018 (r = 0.08, *p* = 0.455 and r = 0.21, *p* = 0.058, resp.). Furthermore, ZEN and ENN B were significantly correlated in 2016 and 2017 (r = 0.35, *p* < 0.001 and r = 0.37, *p* < 0.001, resp.). FUMs were not correlated with any other mycotoxin in any year, except with 15-ADON in 2018 (r = 0.31, *p* = 0.004).

### 3.3. Fusarium spp. DNA in Maize Samples in 2017-2018

Incidence of *F. graminearum*, *F. verticillioides*, *F. culmorum* and *Fusarium* spp. in maize samples in 2017, 2018 and both years combined, is shown in Figure 4; Complete results per sample can be found in Appendix A. In 2017, every maize sample was contaminated with at least one *Fusarium* sp., while in 2018, 36% of the samples were free of *Fusarium* spp. DNA. In both years, *F. verticillioides* was detected most often, with a prevalence of 99% in 2017 and 54% in 2018. *F. graminearum* and *F. culmorum* were detected in 90% and 85% of the maize samples in 2017, and 43% and 51% in 2018, resp.

### 3.4. Correlations between Mycotoxin Concentrations and Fusarium spp. DNA

Using qPCR analysis, we could calculate correlations between mycotoxin concentrations and *Fusarium* spp. DNA on maize fields, and interspecies correlations between *Fusarium* species (Figure 5). Rather weak but significant correlations were found between the amount of *F. graminearum* DNA and *F. culmorum* DNA (r = 0.21, *p* = 0.009) and *F. verticillioides* DNA and *F. culmorum* DNA (r = 0.19, *p* = 0.024). A strong significant correlation was found between DON+ and *F. graminearum* DNA (r = 0.53, *p* < 0.001). Both *F. graminearum* and *F. culmorum* were significantly correlated with higher concentrations of NIV (r = 0.35, *p* < 0.001 and r = 0.36, *p* < 0.001, resp.). Furthermore, *F. verticillioides* DNA was positively correlated with FUM (r = 0.20, *p* < 0.016), but an even stronger correlation was found between FUM and *F. graminearum* (r = 0.27, *p* < 0.001). However, the latter correlation is based primarily on one data point with a high concentration of FUM and a high *F. graminearum* DNA content. When removed from the dataset, the resulting correlation is no longer significant (r = −0.03, *p* = 0.707). Similarly, when removing two outliers from the dataset with a very high FUM content, the correlation between *F. verticillioides* DNA and FUM becomes more profound (r = 0.45, *p* < 0.001). Lastly, when eliminating one outlier from the *F. culmorum* and *F. verticillioides* data, the correlation becomes non-existent (r = −0.06, *p* = 0.482) (See Figure A4). Other correlations were less dependent upon outliers. Splitting the data per year yields similar results as the combined data (Figure A5 and Figure A6).

## 4. Discussion

In our survey, NIV was the most prevalent mycotoxin. Only one out of 257 samples was free of NIV. Concentrations went as high as 6776.3 µg/kg. DON and its derivates 3-ADON and 15-ADON, described together as DON+, were present in 86.8% of the samples. ZEN was found in 49.8% of the samples. Binder et al. [37] took samples of several feedstuffs in Europe and Asia. For maize in Europe, they found DON in 81% of the samples, ZEN in 63%, FUM in 56% and AFB1 in 21%. NIV was not tested. Eckard et al. [24] sampled 20 fields of silage maize in Switzerland for one year. They found DON in every sample, with concentrations up to 2990 µg/kg. ZEN was found in 79% and NIV in 42% of the samples. Goertz et al. [22] sampled maize ears in Germany for two years, and found that incidence and concentrations differed between years. ZEN, NIV and DON and its derivates were detected more frequently and in higher concentrations in a temperate year than in a hot and warm year, while FUMs were only detected in the latter. Van Asselt et al. [38] found that only a quarter of the sampled maize ears in the Netherlands were contaminated with mycotoxins, but 84% of those contained NIV, with concentrations up to 1671 µg/kg. Kosicki et al. [39] found ZEN, DON and NIV in 92%, 89% and 77% of Polish maize ear samples between 2011 and 2014. FUMs were detected in 58% of the samples.

Overall, the results of our survey of Flemish maize are in line with previous research, although the overwhelming incidence and concentrations of NIV have not been described before. NIV is often overlooked when analyzing for mycotoxin contamination.

NIV-producing populations of *F. graminearum* and *F. culmorum* are emerging however [81,82,83], possibly due to the increased use of wheat in a rotation with maize [44]. Furthermore, some reports state NIV may be even more toxic than DON and other trichothecene mycotoxins [84,85,86,87]. More than 24% of the samples in our survey contained NIV concentrations higher than 1000 µg/kg, and 12 samples (4.7%) even exceeded 2000 µg/kg, the EU guidance value for DON. Our research shows that NIV is present in nearly every maize field in Flanders, and often in high concentrations. This mycotoxin should therefore always be included in analyses, especially in Central and North Europe.

Multi-mycotoxin contamination was very common in our survey. Only one sample contained none of the 22 analyzed mycotoxins. 46.7% of maize samples were contaminated with five or more mycotoxins, and two samples in 2018 were even contaminated with 10 different mycotoxins. Schollenberger et al. [62] found up to 12 trichothecenes in one sample, Drejer Storm et al. [48] found up to seven mycotoxins in one sample, and Streit et al. [67] found that up to 69 secondary metabolites including mycotoxins may co-occur in one sample. This multi-mycotoxin contamination is not covered in the current EU regulations. In our survey, 2% of the samples exceeded the EU guidance value for DON, and 7.8% for ZEN. None of the samples exceeded the guidance values or maximum levels for FUM, AFB1 or T2. However, other mycotoxins (e.g., NIV) are not included in the EU regulations, and as mentioned before, multi-mycotoxin contamination and possible synergistic effects are also not included. The sample with the highest overall mycotoxin concentration in our survey contained seven different mycotoxins, i.e. NIV, DON, FX, 3-ADON, 15-ADON, DAS and ZEN, with a total mycotoxin load of 13,474.6 µg/kg. This sample only exceeded the EU regulation for DON, being 2000 µg/kg. But one could assume that its toxicity will be far higher than that of a sample containing only DON in a concentration above 2000 µg/kg. Synergistic or additive toxic effects of a combined mycotoxin contamination have been demonstrated in previous literature, especially with mycotoxins that share a similar chemical structure or are produced by the same fungal species [68,70,71,72,88]. The EU regulations should therefore be re-evaluated and expanded in the future to account for multi-mycotoxin contamination [2].

As expected, the concentrations of DON and its derivates 3-ADON and 15-ADON were strongly positively correlated [66]. Similarly, the fumonisins FB1, FB2 and FB3, were strongly correlated. Other significant positive correlations were found between NIV; and DON (and its derivates), ZEN and ENN B. NIV is known to be primarily produced by *F. culmorum* in temperate regions, while DON is mainly produced by *F. graminearum* [35,77,89,90]. Since DON and NIV are positively correlated, the amount of *F. graminearum* and *F. culmorum* DNA were expected to be positively correlated as well. This was indeed the case, although the correlation was not particularly strong (r = 0.21).

FUMs were not correlated with any other mycotoxin. Likewise, the main fumonisin producing *Fusarium* species, *F. verticillioides*, was not correlated with *F. graminearum*. The correlation between *F. verticillioides* and *F. culmorum* was significant (r = 0.19), despite the absence of a correlation between NIV and FUM. However, as explained earlier in the Results section, some outliers may have skewed the data. In this case, omitting one outlier from the dataset effaced the corresponding correlation between *F. verticillioides* and *F. culmorum* (r = -0.06). Similarly, removing one outlier made the unexpected correlation between *F. graminearum* and FUM non-significant (r = -0.03), and removing two outliers made the expected but rather weak correlation between *F. verticillioides* and FUM more profound (r = 0.45). Other correlations were stable and less dependent upon outliers. With these adjustments, we could conclude that the main fumonisin-producer *F. verticillioides* is positively correlated with FUM; the main NIV-producer *F. culmorum* is correlated with NIV; and the main DON-producer *F. graminearum* is correlated with DON+. The latter relation could be demonstrated anecdotally, because the sample with the highest DON+ concentration also had the highest levels of *F. graminearum* DNA. These results are in accordance with the previous literature [22,24,83]. Other researchers have found a correlation between DON and ZEN [39,40,61,91], which was not the case in our survey, except in 2017 (Figure A2).

*Fusarium graminearum* and *F. culmorum* share a positive correlation, meaning that they can co-exist and produce mycotoxins on the same plant. On the other hand, *F. verticillioides* is not correlated with *F. graminearum* nor with *F. culmorum*.

This could be caused by differing optimal growing conditions, since *F. verticillioides* prefers warm temperatures and dry conditions, while *F. graminearum* and *F. culmorum* both prefer colder and wetter conditions [19,23]. Moreover, the co-occurrence of different fungal species on the same plant may have a significant impact on fungal development and mycotoxin production [92,93,94]. Indeed, most plant diseases are caused by a complex of species rather than by a single species, which may lead to synergistic effects [95]. Previous research has shown that *F. graminearum* and *F. verticillioides* may co-occur and produce mycotoxins on the same plant when infected artificially, but the type of interactions may differ depending on the weather conditions [25,26,27,29,96]. FUM production is mainly reduced when *F. graminearum* and *F. verticillioides* are co-inoculated, whereas DON production is increased; ZEN production is not affected [26]. When co-inoculated with *Aspergillus parasiticus*, ZEN and DON production by *F. graminearum* is not infected, while AFB1 production by *A. parasiticus* is significantly reduced [97]. Furthermore, a high amount of fungal inoculum does not necessarily lead to higher mycotoxin concentrations [98]. These effects of fungal co-occurrence may explain why *F. graminearum* and *F. verticillioides* are not correlated in our survey, and why certain expected correlations between fungal species and/or mycotoxins have not been observed.

There was a clear year-to-year difference in the observed mycotoxin incidences and concentrations and the presence of *Fusarium* spp. DNA, related to changes in the weather conditions. This has been observed multiple times in past literature [22,40,58,99]. A summary of the weather conditions of each year (2016–2018) can be found in Table 2. 2016 was a year with high precipitation, especially in June, and a high relative humidity (RH). 2017 had less rainfall and less radiation, but similar temperatures. 2018 was an extremely dry year, with only 241 mm of precipitation during the growing season, and the highest temperatures ever recorded in Belgium, up to 41.8 °C on the 25^th^ of July [100]. These extreme, dry and warm temperatures led to a number of different observations: More diversely-contaminated samples, but a lower median mycotoxin load per sample; A reduction of the incidence and concentrations of DON and its derivates; more samples that were highly contaminated with ZEN, and thereby exceeded EU guidance values; more incidence of *Alternaria* mycotoxins AOH and AME; and most remarkably, a strong increase in the incidence and concentrations of FUMs. 61.2% of maize samples were contaminated with FUMs in 2018, with a concentration of up to 6293.5 µg/kg, versus 19.8% in 2017 and only 2.5% in 2016. Contrastingly, the incidence of *F. verticillioides* did not rise, but was lower compared to 2017 (99% and 54%, resp.). Since mycotoxin production is influenced by temperature and water levels [23,26,101,102], the specific growing conditions in 2018 could have reduced *F. verticillioides* infection but induced FUM production. In general, less maize samples were contaminated with *Fusarium* spp. in 2018 compared to 2017 (100% and 64%, resp.), with *F. verticillioides* being the most prevalent species in both years. Scauflaire et al. [21] found that in maize ears and stalks in Wallonia, Belgium, *F. graminearum* was the predominant species, while *F. verticillioides* occurred only sporadically. The same conclusions were drawn in Switzerland [61] and the UK [44]. The dissimilar results of our survey compared to these studies could be explained by the abnormal weather conditions in Belgium in 2017 and 2018, causing a shift in the fungal populations. *F. verticillioides* infection and, correspondingly, FUM production is higher in warm and dry years [22,23,40,103]. Many maize fields in 2018 were of very low quality and were harvested with little to no cobs developed, possibly explaining the lower general incidence of *Fusarium* spp. in that year. Furthermore, *Fusarium* spp. generally infect a plant in a species complex [19]. Only three *Fusarium* spp. were included in our qPCR analysis. It is possible that other species were present as well, and produced mycotoxins of their own. In the previous literature, 11 to 23 different *Fusarium* species were isolated from maize fields in Belgium [21], the UK [44], Switzerland [24,61], Germany [22] and the Netherlands [38]. Possibly, infections by *F. poae* (NIV, DAS), *F. avenaceum* (ENN B), *F. proliferatum* (FUM), *F. crookwellense* (NIV, ZEN) or other *Fusarium* species occurring in Belgium [21,74] could explain the incidence of certain related mycotoxins [19,104].

## 5. Conclusions

In conclusion, this 3-year study has demonstrated the shifting mycotoxin load in silage maize fields at harvest due to changing weather conditions, possibly induced by climate change. Fumonisins, produced by *F. verticillioides*, which is more prevalent in tropical climates, were detected sporadically in Flanders in wet and cold years, but were found far more frequent during dry and hot years. Nivalenol was found in all but one of the samples, across all three years, making it the most stable and widespread mycotoxin. Concentrations went as high as 6776 µg/kg. Aflatoxins were not found, but *Aspergillus* spp. grow at similar conditions as *F. verticillioides*, so these mycotoxins should not be overlooked in future surveys. In order to monitor the effect of climate change on these changing weather conditions and on subsequent mycotoxin production, a yearly sampling should be continued.

The next step will be to identify the underlying cultivation, environmental and climatic factors that influence mycotoxin contamination in the field, and to create a prediction model for farmers based on these data. Ultimately, this research could help reduce mycotoxin contamination in silage maize and reduce mycotoxicosis in dairy cattle.

## Figures and Tables

**Figure 1 microorganisms-07-00571-f001:**
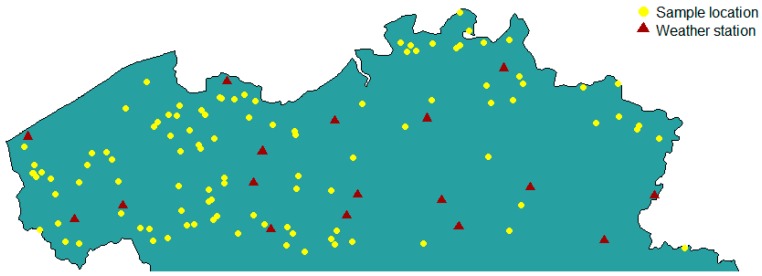
Location of the 106 dairy farms and 17 weather stations in Flanders, Belgium.

**Figure 2 microorganisms-07-00571-f002:**
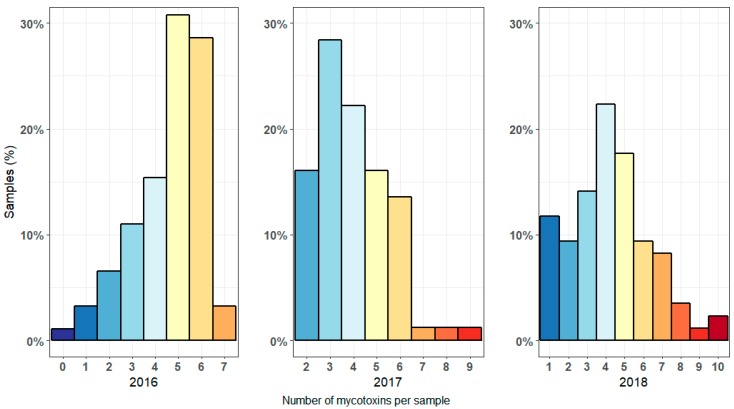
The relative number of maize samples contaminated with a certain number of different mycotoxins for 2016, 2017 and 2018.

**Figure 3 microorganisms-07-00571-f003:**
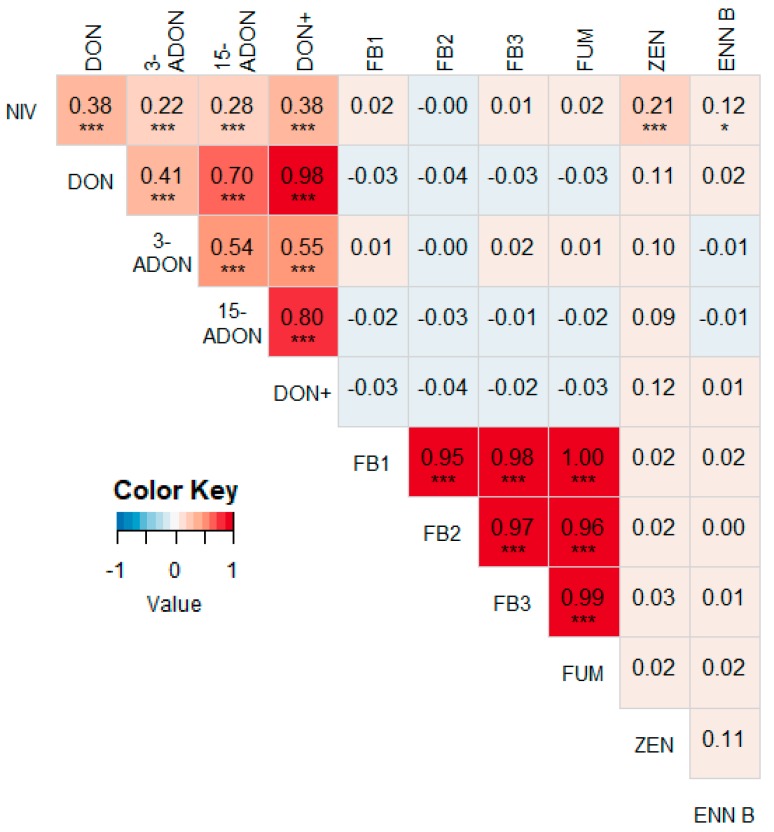
Heat map based on the pairwise Pearson correlation coefficients between the measured mycotoxin concentrations from 2016–2018. A darker blue color indicates a stronger negative correlation, a darker red color indicates a stronger positive correlation. Significant correlations are indicated with asterisks (* *p* < 0.05, *** *p* < 0.01). DON+ = the sum of the concentrations of DON, 3-ADON and 15-ADON. FUM = the sum of the concentrations of FB1, FB2 and FB3.

**Figure 4 microorganisms-07-00571-f004:**
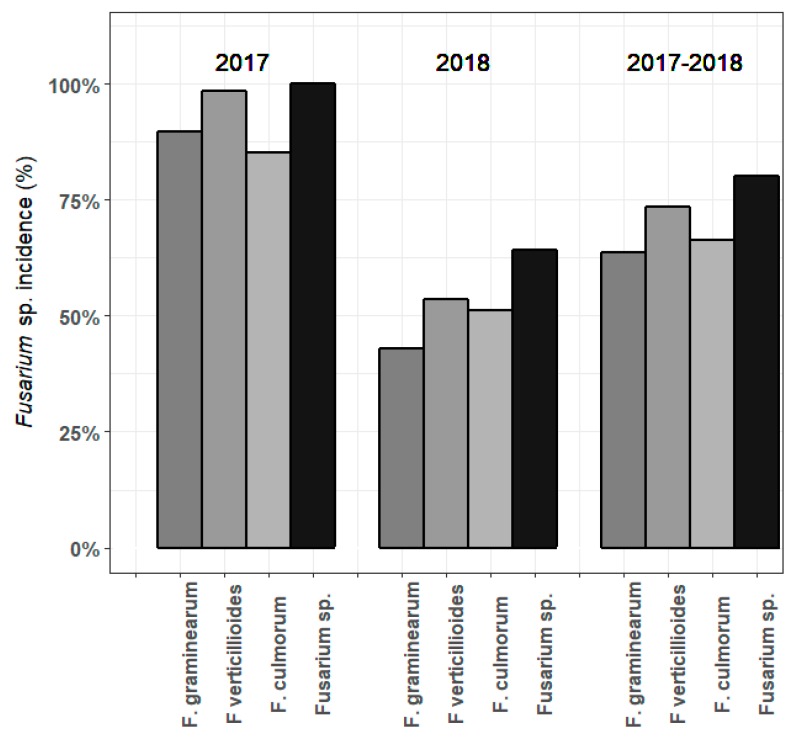
Incidence of *F. graminearum*, *F. verticillioides*, *F. culmorum* and *Fusarium* spp. in general, in samples of harvested maize in 2017, 2018 and in both years combined.

**Figure 5 microorganisms-07-00571-f005:**
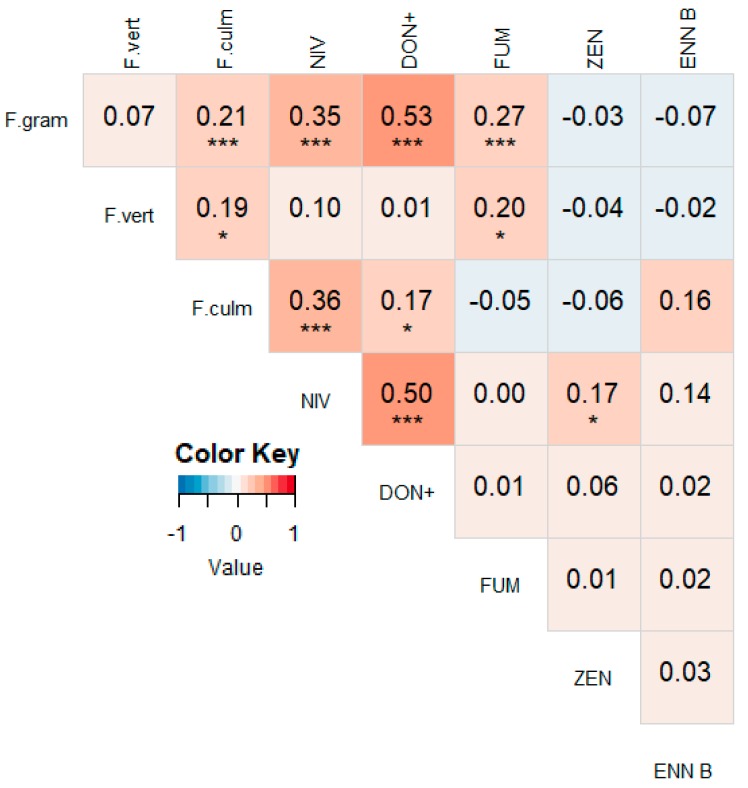
Heat map based on the pairwise Pearson correlation coefficients between measured mycotoxin concentrations and DNA of *F. graminearum*, *F. verticillioides* and *F. culmorum* from 2017–2018. A darker blue color indicates a stronger negative correlation, a darker red color indicates a stronger positive correlation. Significant correlations are indicated with asterisks (* *p* < 0.05, *** *p* < 0.01). DON+ = the sum of the concentrations of DON, 3-ADON and 15-ADON. FUM = the sum of the concentrations of FB1, FB2 and FB3.

**Table 1 microorganisms-07-00571-t001:** Mycotoxin contamination detected in maize samples at harvest in Flanders, Belgium, from 2016 till 2018.

	Positive Samples (%)				Mean Concentration ^a^ (µg/kg)				Median Concentration (µg/kg)				Max. Concentration (µg/kg)				Samples Exceeding EU Recommendation (%)^b^			
	2016	2017	2018	2016–2018	2016	2017	2018	2016–2018	2016	2017	2018	2016–2018	2016	2017	2018	2016-2018	2016	2017	2018	2016–2018
n samples	91	81	85	257	91	81	85	257	91	81	85	257	91	81	95	257	91	81	95	257
NIV	98.9	100	98.8	99.2	650.7	719.0	881.9	748.7	527.5	460.6	782.1	587.1	2368.2	6776.3	2409.5	6776.3				
DON	92.3	100	64.7	85.6	449.0	557.5	186.5	396.4	263.1	337.4	121.3	215.3	2777.4	5322.4	2110.5	5322.4	2.2	3.7	1.0	2.3
3-ADON	78.0	29.6	15.3	42.0	53.5	36.3	23.3	38.1	43.2	0	0	0	297.0	380.3	1046.8	1046.8				
15-ADON	64.8	51.3	12.9	43.4	95.0	81.2	15.2	64.5	71.3	17.7	0	0	819.3	769.2	248.6	819.3				
DON+^c^	95.6	100	64.7	86.8	597.5	675.0	225.0	498.7	376.9	406.7	130.3	261.5	3050.1	6471.9	2110.5	6471.9				
ZEN	64.8	40.7	42.4	49.8	100.5	158.5	175.5	159.7	71.2	0	0	0	1085.6	1411.9	2791.6	2791.6	1.1	8.6	12.6	7.8
ENN B	42.9	18.5	45.9	36.2	133.1	77.7	180.3	149.5	56.2	27.5	70.7	46.2	1375.1	1041.9	1984.9	1984.9				
AOH	n.d.	3.7	9.4	4.3	n.d.	1.4	6.5	2.6	n.d.	0	0	0	n.d.	49.1	208.6	208.6				
AME	2.2	3.7	10.6	5.4	0.8	11.5	19.8	10.5	0	0	0	0	49.1	370.6	452.8	452.8				
FB1^d^	2.5	19.8	61.2	28.6	1.5	61.1	247.4	106.5	0	0	54.0	0	70.2	1362.9	4414.9	4414.9	0	0	0	0
FB2^d^	n.d.	4.9	24.7	10.2	n.d.	9.0	61.6	24.4	n.d.	0	0	0	n.d.	412.6	1427.4	1427.4	0	0	0	0
FB3^d^	n.d.	7.4	18.8	9.0	n.d.	3.4	18.0	7.4	n.d.	0	0	0	n.d.	90.5	451.2	451.2				
FUM^c^	2.5	19.8	61.2	28.6	1.3	73.6	327.0	131.8	0	0	58.7	0	70.2	1782.8	6293.5	6293.5				
DAS	11.0	8.6	5.9	8.6	0.3	0.3	0.4	0.4	0	0	0	0	6.1	10.3	14.9	14.9				
FX	n.d.	7.4	2.4	3.1	n.d.	14.2	2.7	5.4	n.d.	0	0	0	n.d.	223.6	161.6	223.6				
T2	1.1	n.d.	8.2	3.1	0.2	n.d.	6.2	2.1	0	n.d.	0	0	16.8	n.d.	121.6	121.6	0	0	0	0
STERIG	1.1	n.d.	1.2	0.8	0.2	n.d.	2.6	0.9	0	n.d.	0	0	15.1	n.d.	73.3	204.8				
ROQ-C^d^	n.d.	2.5	2.9	1.7	n.d.	0.6	0.6	0.4	n.d.	0	0	0	n.d.	30.4	24.6	30.4				
TOTAL^c^	98.9	100	100	100	1485.1	1729.9	1877.4	1692.0	1309.6	1088.2	1596.1	1309.6	4153.4	13747.6	8309.0	13747.6				

n.d.: Not detected. a: Arithmetic mean. b: EU regulations: 2000 µg/kg for DON (complementary and complete feedstuffs for calves (< 4 months)); 500 µg/kg for ZEN complementary and complete feedstuffs for calves and dairy cattle; 20,000 µg/kg for FB1+FB2 (calves (< 4 months)); 250 µg/kg for T2 (compound feed) (European Commission, 2006, 2013). c: DON+ = the sum of the incidence/concentrations of DON, 3-ADON and 15-ADON; FUM = the sum of the incidence/concentrations of FB1, FB2 and FB3; TOTAL = The sum of the incidence/concentrations of all detected mycotoxins. d: In 2016, only 79 samples were analyzed for FB1, FB2 and FB3. In 2018, only 68 samples were analyzed for ROQ-C.

**Table 2 microorganisms-07-00571-t002:** Weather parameters during the 2016–2018 maize growing seasons in Flanders, Belgium. The growing season start date is based on the first maize field in our database being sown; the end date is based on the last maize field being harvested. Mean and range values are based on daily weather measurements from 17 weather stations across Flanders (Figure 1).

Year	Growing Season	Rainfall (mm)		Relative Humidity (%)		Average Temperature (°C)		Total Daily Radiation (W/m²)	
		Mean	Range (Min. - Max.)	Mean	Range (Min. - Max.)	Mean	Range (Min. - Max.)	Mean	Range (Min. - Max.)
2016	20.04–26.10 (189 days)	423	283–610	80.5	72.8–86.3	15.7	15.1–16.7	3839	3697–4024
2017	10.04–28.10 (201 days)	344	186–541	78.6	70.3–85.3	15.7	15.1–16.2	3389	2600–3902
2018	19.04–14.10 (178 days)	241	155–335	74.2	69.6–84.7	17.5	16.0–18.3	4294	4109–4710

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
