# Peer review of "Mycotoxins in Flanders’ Fields: Occurrence and Correlations with Fusarium Species in Whole-Plant Harvested Maize"

_microorganisms, 2019, doi:10.3390/microorganisms7110571_

Round 1

Reviewer 1 Report

The manuscript is well written and contain novel information worthy of review.  My only criticism of the manuscript is the choice of PCR assays to identify the Fusarium species present.  Based on the mycotoxin profile it would have been very useful to include more known nivalenol and fumonisin producers.  This is of particular interest for nivalenol as high levels were detected and not clear what species is responsible (although correlation does not prove a cause and effect as seen in the results presented here).  Also the authors can not claim to see impact of climate change has by comparing results over three years (two wet and one dry).  Differences again maybe purely incidental and would need several years data before can claim to identify climate effects.

Minor comments:

L36 include forage and silage in keywords

Table 1 EU Recommendation not Regulation for toxins present in feedstuffs

Figure 2 unnecessary, can be easily described in the text

Discussion needs to include high nivalenol levels detected

Conclusion needs removal of climate change claim in first sentence

Author Response

Mycotoxins in Flanders’ fields: Occurrence and correlations with Fusarium species in whole-plant harvested maize

RESPONSE TO REVIEWER 1

Dear Ms. Xi, Dr. Stępień, and reviewer 1,

Thank you for your remarks and thoughts on the paper. We have reviewed them carefully and responded to each comment separately in the text below.

Kind regards,

The authors

P.S. The references in the original manuscript were processed using Mendeley. In the manuscript sent after the revisions, all references were transformed to plain text form. To simplify the addition of extra references to the manuscript, the original manuscript was used for the resubmission, and the changes that have been made to the lay-out during revision were readressed.

Q1: “The manuscript is well written and contains novel information worthy of review.  My only criticism of the manuscript is the choice of PCR assays to identify the Fusarium species present.  Based on the mycotoxin profile it would have been very useful to include more known nivalenol and fumonisin producers.  This is of particular interest for nivalenol as high levels were detected and not clear what species is responsible (although correlation does not prove a cause and effect as seen in the results presented here).”

A1: We agree with reviewer that the list of Fusarium species could have been extended. Due to the rather high number of samples, we decided to limit the number of species to three, based on the known fungal species composition in temperate climates and known to be omnipresent in Belgium (See addition in text, P5 L9-11 and additional reference (Ref. 74 P24 L7)). However, more fungal species could have been included. Especially F. poae (as a NIV-producer), F. avenaceum (as an ENN B producer) and F. proliferatum (as a fumonisin producer) could have been interesting additions. Due to time constraints it was not possible to repeat these qPCR assays with the additional species, therefore we have added more nuance concerning the limited number of analyzed species to the text (P12, L33-38).

Q2: “Also the authors cannot claim to see impact of climate change has by comparing results over three years (two wet and one dry).  Differences again maybe purely incidental and would need several years data before can claim to identify climate effects.”

A2: We agree with the reviewer that the impact of climate change on these three-year results was described too distinctly in the discussion and conclusion. Although there is an indication that differences in climatic conditions in the different growing seasons had an impact on the changing weather conditions in Flanders, especially considering the extreme and record-breaking temperatures in 2018, it is too premature to put this forward as the main driver for a shift in the mycotoxin load in maize samples. Therefore, some changes were made to the text.

First alinea of discussion (P10), including impact of climate change, was removed according to recommendation of Reviewer 2; Some adaptations were made in Conclusions (P14, L3 & L9-10).

“Minor comments:”

Q3: “L36 include forage and silage in keywords”

A3: ‘Forage’ and ‘silage’ have been added to the keywords. Since these additions would exceed the maximum limit of 10 keywords, ‘LC-MS/MS’ and ‘qPCR’ have been omitted (P1, L34).

Q4: “Table 1 EU Recommendation not Regulation for toxins present in feedstuffs”

A4: Has been adapted (P6, Table 1).

Q5: “Figure 2 unnecessary, can be easily described in the text”

A5: In accordance to this comment and to a recommendation by reviewer 2, we have replaced Figure 2 by three figures showing the data of each year separately (P7). Interpretation of the overall results has been adapted (P7, L11-13) and additional interpretation of the yearly results has been added in text (P7, L14-19 and P12, L13-14)

Q6: “Discussion needs to include high nivalenol levels detected”

A6: Discussion about NIV has been expanded in text (P11, L1-8 and P14, L6-7) and an extra reference to the higher toxicity of NIV compared to DON (ref. 87, P24, L39)

Q7: “Conclusion needs removal of climate change claim in first sentence”

A7: Has been nuanced in text (P14, L3 & L9-10)

Reviewer 2 Report

In their manuscript untitled « Mycotoxins in Flanders’ fields: Occurrence and correlations with Fusarium species in whole-plant harvested maize », the authors described the multi-mycotoxins contamination of 257 maize samples taken from fields across Flanders, Belgium, over the course of three years (2016-2018) and researched the link between their level in samples and the contamination by the three most common Fusarium species.

Since the incidence and concentration have been presented over the years, it will be interesting to also have the number of mycotoxins per sample over the years to see if there is a change over the time. I will suggest turning Figure 2 of a graph into 3 graphs, one for each year and comment this evolution in the manuscript.

The first paragraph of the discussion contains information already mentioned in the introduction. Because this information justifies the objective of the study, its place will fit better in the introduction. I will suggest deleting it and starting directly with the main results of your study.

In the discussion section, p11 line 31-32: The authors can support this affirmation by citing the increasing number of publications showing the higher toxic effect of mycotoxin mixture on human and/or animal cells (as for example doi: 10.3390/toxins9110337: Primary and Immortalized Human Respiratory Cells Display Different Patterns of Cytotoxicity and Cytokine Release upon Exposure to Deoxynivalenol, Nivalenol and Fusarenon-X. Ferreira Lopes S, Vacher G, Ciarlo E, Savova-Bianchi D, Roger T, Niculita-Hirzel H. Toxins (Basel). 2017 Oct 25;9(11). pii: E337.)

Small comment: I suppose that all refrigerators were at the same temperature -20°C or -18°C. No?  If so, please homogenise through the material and methods section p3 line 29, p4 line 7, 9 and 14.

Author Response

Mycotoxins in Flanders’ fields: Occurrence and correlations with Fusarium species in whole-plant harvested maize

RESPONSE TO REVIEWER 2

Dear Ms. Xi, Dr. Stępień, and reviewer 2,

Thank you for your remarks and thoughts on the paper. We have reviewed them carefully and responded to each comment separately in the text below.

Kind regards,

The authors

P.S. The references in the original manuscript were processed using Mendeley. In the manuscript sent after the revisions, all references were transformed to plain text form. To simplify the addition of extra references to the manuscript, the original manuscript was used for the resubmission, and the changes that have been made to the lay-out during revision were readressed.

Q1: “In their manuscript untitled « Mycotoxins in Flanders’ fields: Occurrence and correlations with Fusarium species in whole-plant harvested maize », the authors described the multi-mycotoxins contamination of 257 maize samples taken from fields across Flanders, Belgium, over the course of three years (2016-2018) and researched the link between their level in samples and the contamination by the three most common Fusarium species.

Since the incidence and concentration have been presented over the years, it will be interesting to also have the number of mycotoxins per sample over the years to see if there is a change over the time. I will suggest turning Figure 2 of a graph into 3 graphs, one for each year and comment this evolution in the manuscript.”

A1: Figure 2 (P7) has been replaced by three figures representing the data from 2016, 2017 and 2018 separately, as recommended by the reviewer. Interpretation of the overall results has been adapted (P7, L11-13) and additional interpretation of the yearly results has been added in text (P7, L14-19 and P12, L13-14).

Q2: “The first paragraph of the discussion contains information already mentioned in the introduction. Because this information justifies the objective of the study, its place will fit better in the introduction. I will suggest deleting it and starting directly with the main results of your study.”

A2: First paragraph of the discussion has been removed (P10), some lines were moved to Materials and Methods (P5, L9-11).

Q3: “In the discussion section, p11 line 31-32: The authors can support this affirmation by citing the increasing number of publications showing the higher toxic effect of mycotoxin mixture on human and/or animal cells (as for example doi: 10.3390/toxins9110337: Primary and Immortalized Human Respiratory Cells Display Different Patterns of Cytotoxicity and Cytokine Release upon Exposure to Deoxynivalenol, Nivalenol and Fusarenon-X. Ferreira Lopes S, Vacher G, Ciarlo E, Savova-Bianchi D, Roger T, Niculita-Hirzel H. Toxins (Basel). 2017 Oct 25;9(11). pii: E337.)”

A3: Some additions have been made in text (P11, L23-26), and several references have been added, including the one recommended by the reviewer (Ref. 88, P24, L41).

Q4: “Small comment: I suppose that all refrigerators were at the same temperature -20°C or -18°C. No?  If so, please homogenise through the material and methods section p3 line 29, p4 line 7, 9 and 14.”

A4: Has been homogenized to -20 °C (P3 L27; P4 L5, L7 and L12).

Reviewer 3 Report

Mycotoxins, secondary metabolites of different mold species contaminate the feed ingredients, causing economic losses due to the lower performance and reduced fertility of monogastric and ruminant species. The changing climate and variation in weather conditions year-to-year may affect the mold population on plants resulting alterations in mycotoxin incidence and concentrations in feedstuffs.
In their 3-year study authors investigated the natural mycotoxin content in harvested maize plants for ensiling in Flanders (Belgium), and the link between the different mycotoxins and the mycotoxinogenic Fusarium species.
Although there are quite similar studies available from Europe (from Germany, Switzerland, etc.), the strength of this study is that the authors analyzed a lot of samples (257) for 22 different mycotoxins, combined with DNA analysis of the most prevalent Fusarium species by qPCR.
The survey is easy to read and the results are clearly presented, so that highly understandable; the experimental design was well structured and the conclusions are supported by data.
My question for the mycotoxin analyses:
Several mycotoxins and their metabolites were measured from the samples, but why was not HT-2 toxin detected (although the EU regulation handles T-2 toxin and HT-2 toxin together)?

Some comments to the manuscript:
At P2 L4: I advice to highlight that the mycotoxin producing fungi are different molds.
At P2 L28: I advice to rewrite this paragraph, as NIV, T2, DAS are also trichothecenes, while ZEN,
and FUM are not.
At P2 L30: use the abbreviation FUM, instead of FB for fumonisins.
At P3 L27 and 29: use ca. instead of +/-
At P4 L19: what is the right chemical name of ZAN (which is a structural analogue of ZEN)?
At P4 L41: correct abbreviation: AFB2
At P6 (Table 1): Why is the % of positive samples in 2016 different for DON and DON+?
At P6 (Table 1) L5: I advice to use for DON+ and FUM
the sum of the incidence/concentrations of …

Figure A1 (P15) does not contain rows and columns for FB2 and FB3. If FB2 and FB3 were not detected in year 2016, I advice to use n.d. (not detected) instead of 0 for their incidence and concentrations in Table 1 (P6).

In every Figure use the same abbreviation (ENN B) for enniatin B what you wrote in the text corpus.
At P11 L28: I advice to use:
positive synergistic effects are also not included.
At P11 L28: I advice to use:
NIV; and
At P12 L5: correct: Figure A2
P 14 L5: use Nivalenol instead of NIV, as you did not use the abbreviated form for fumonisins and aflatoxins in the same paragraph.

In References correct the style of the followings:
Ref. 11; Ref. 14, Ref. 27, Ref. 28, Ref. 47, Ref. 51, Ref. 69, Ref. 88

At Ref. 43 B1 is correct
At Ref. 53 European is correct

The Figures in Appendix have the same numbering as the Figures in text.
At P15, P16, P17, P18, P19 use Figure A1-A5.

P17 L3: these correlations are for year 2018, not for 2017

Author Response

Mycotoxins in Flanders’ fields: Occurrence and correlations with Fusarium species in whole-plant harvested maize

RESPONSE TO REVIEWER 3

Dear Ms. Xie, Dr. Stępień, and reviewer 3,

Thank you for your remarks and thoughts on the paper. We have reviewed them carefully and responded to each comment separately in the text below.

Kind regards,

The authors

P.S. The references in the original manuscript were processed using Mendeley. In the manuscript sent after the revisions, all references were transformed to plain text form. To simplify the addition of extra references to the manuscript, the original manuscript was used for the resubmission, and the changes that have been made to the lay-out during revision were readressed.

Q1: “Mycotoxins, secondary metabolites of different mold species contaminate the feed ingredients, causing economic losses due to the lower performance and reduced fertility of monogastric and ruminant species. The changing climate and variation in weather conditions year-to-year may affect the mold population on plants resulting alterations in mycotoxin incidence and concentrations in feedstuffs.

In their 3-year study authors investigated the natural mycotoxin content in harvested maize plants for ensiling in Flanders (Belgium), and the link between the different mycotoxins and the mycotoxinogenic Fusarium species.

Although there are quite similar studies available from Europe (from Germany, Switzerland, etc.), the strength of this study is that the authors analyzed a lot of samples (257) for 22 different mycotoxins, combined with DNA analysis of the most prevalent Fusarium species by qPCR.

The survey is easy to read and the results are clearly presented, so that highly understandable; the experimental design was well structured and the conclusions are supported by data.

My question for the mycotoxin analyses:

Several mycotoxins and their metabolites were measured from the samples, but why was not HT-2 toxin detected (although the EU regulation handles T-2 toxin and HT-2 toxin together)?”

A1: In the first year of sampling, HT-2 was included in the set of mycotoxins to be analyzed with LC-MS/MS. However, it was difficult to detect HT-2 and to draw a reliable calibration curve to quantify this component. Presumably HT-2 was masked by the matrix (dried maize). Since only traces were detected, and since the related mycotoxin T-2 was only detected in a small number of samples, we omitted HT-2 from further analysis.

“Some comments to the manuscript:”

Q2: “At P2 L4: I advice to highlight that the mycotoxin producing fungi are different molds.”

A2: The adjective ‘moldy’ has been added before ‘fungi’ to express this specification (P2, L2).

Q3: “At P2 L28: I advice to rewrite this paragraph, as NIV, T2, DAS are also trichothecenes, while ZEN, and FUM are not.”

A3: Adapted in text (P2, L26-27)

Q4: “At P2 L30: use the abbreviation FUM, instead of FB for fumonisins.”

A4: Adapted in text (P2, L28).

Q5: “At P3 L27 and 29: use ca. instead of +/-“

A5: Adapted in text (P3, L25 & L27).

Q6: “At P4 L19: what is the right chemical name of ZAN (which is a structural analogue of ZEN)?”

A6: The chemical name of ZAN (zearalanone) has been added to the text (P4, L17).

Q7: “At P4 L41: correct abbreviation: AFB2”

A7: Adapted in text (P5, L42).

Q8: “At P6 (Table 1): Why is the % of positive samples in 2016 different for DON and DON+?”

A8: There were three samples in 2016 which contained a detectable concentration of 3-ADON (of which one also contained 15-ADON), but did not contain a detectable amount of DON. Therefore these samples are included in the incidence results for DON+, but not for DON. This occurred only in these three samples in 2016, none in 2017 nor in 2018.

Q9: “At P6 (Table 1) L5: I advice to use for DON+ and FUM the sum of the incidence/concentrations of …”

A9: Adapted in text (P6, table 1).

Q10: “Figure A1 (P15) does not contain rows and columns for FB2 and FB3. If FB2 and FB3 were not detected in year 2016, I advice to use n.d. (not detected) instead of 0 for their incidence and concentrations in Table 1 (P6).”

A10: Adapted in Table 1 (P6). FB2 and FB3 were indeed not detected in 2016, and were therefore omitted from Figure A1 (P15).

Q11: “In every Figure use the same abbreviation (ENN B) for enniatin B what you wrote in the text corpus.”

A11: Adapted in Figures 3 (P8), 5 (P10), A1 (P15), A2 (P16), A3 (P17), A4 (P18), A5 (P19) and A6 (P20).

Q12: “At P11 L28: I advice to use: positive synergistic effects are also not included.”

A12: Adapted in text (P11, L18-19).

Q13: “At P11 L28: I advice to use: NIV; and”

A13: Adapted in text (P11, L30-31).

Q14: “At P12 L5: correct: Figure A2”

A14: Adapted in text (P11, L50)

Q15: “P 14 L5: use Nivalenol instead of NIV, as you did not use the abbreviated form for fumonisins and aflatoxins in the same paragraph.”

A15: Adapted in text (P14, L6).

Q16: “In References correct the style of the followings:

Ref. 11 ; Ref. 14 , Ref. 27 , Ref. 28 , Ref. 47 , Ref. 51 , Ref. 69 , Ref. 88

At Ref. 43  B1 is correct

At Ref. 53  European is correct”

A16: Adapted in reference list (Ref. 11 P21 L12; Ref. 14 P21 L18; Ref. 27 P21 L49; Ref. 28 P22 L2; Ref. 43 P22 L32; Ref. 48 P22 L43; Ref. 52 P23 L3; Ref. 54 P23 L6; Ref. 70 P23 L46; Ref. 96 P25 L9)

Q17: “The Figures in Appendix have the same numbering as the Figures in text. At P15, P16, P17, P18, P19 use Figure A1-A5.”

A17: Adapted in text (P15-19).

Q18: “P17 L3: these correlations are for year 2018, not for 2017”

A18: Adapted in text (P17, L3).

Round 2

Reviewer 2 Report

The authors answered my previous comments. However, I have one major and two minor comments concerning this second version of the manuscript.

My major comment concern the Discussion

The authors point out the co-occurrence of F. graminearum and F. culmorum but not of F. verticillioides, after they discuss the correlations between DNA and mycotoxins concentrations in the previous two paragraphs. They also discuss the strong impact of climate on mycotoxins production. I will suggest that they complete the discussion page 12 after line 5 with a small paragraph on the importance of fungal interaction (species co-occurrence) on Fusarium development and mycotoxin production. Indeed, mycotoxin production are predicted to be more complex when more than one toxigenic species is present. Such an interaction can explain weak or absent correlations between DNA and mycotoxins concentrations observed in the present study as well as the differences with the other published studies. One review summarizing this concept is that of Ferrigo et al. (Molecules 2016, 21, 627; doi:10.3390/molecules21050627).

Few examples of such impact of fungal interactions are:

Fumonisin production has been found generally reduced in competing interactions, whereas zearalenone was not affected and DON was increased in co-inocultation experiments (Velluti, A.; Marín, S.; Gonzalez, R.; J Ramos, A.; Sanchis, V. Fumonisin B1, zearalenone and deoxynivalenol production by Fusarium moniliforme, F. proliferatum and F. graminearum in mixed cultures on irradiated maize kernels. J. Sci. Food Agric. 2001, 81, 88–94.) It was demonstrated that high levels of F. verticillioides do not necessarily result in high levels of fumonisin contamination (Ferrigo, D.; Raiola, A.; Causin, R. Plant stress and mycotoxin accumulation in maize. Agrochimica 2014, 58, 116–127.) Insensitivity of ZEA and DON producers to competition occurred when F. graminearum was cultivated with Aspergillus parasiticus, and the toxin levels were not modified (Etcheverry, M. Aflatoxin B1, zearalenone and deoxynivalenol production by Aspergillus parasiticus and Fusarium graminearum in interactive cultures on irradiated corn kernels. Mycopathologia 1998, 142, 37–42)

My two minor comments concern:

Page 5 Line10 : Please complete “to cover most Fusarium mycotoxins that were included in the LC-MS/MS analysis” as following “to cover most Fusarium producers of mycotoxins that were included in the LC-MS/MS analysis”

Page 11 Line 45 : Replaced the upper-case by a lower-case for “And” in the sentence :” F. culmorum is correlated with NIV; And the main”

Author Response

Mycotoxins in Flanders’ fields: Occurrence and correlations with Fusarium species in whole-plant harvested maize

RESPONSE TO REVIEWER 2

Dear reviewer 2,

Thank you for your comments on the paper. We have adapted the manuscript accordingly. More details can be found in the text below.

Kind regards,

The authors

Q1: “The authors answered my previous comments. However, I have one major and two minor comments concerning this second version of the manuscript.

My major comment concern the Discussion:

The authors point out the co-occurrence of F. graminearum and F. culmorum but not of F. verticillioides, after they discuss the correlations between DNA and mycotoxins concentrations in the previous two paragraphs. They also discuss the strong impact of climate on mycotoxins production. I will suggest that they complete the discussion page 12 after line 5 with a small paragraph on the importance of fungal interaction (species co-occurrence) on Fusarium development and mycotoxin production. Indeed, mycotoxin production are predicted to be more complex when more than one toxigenic species is present. Such an interaction can explain weak or absent correlations between DNA and mycotoxins concentrations observed in the present study as well as the differences with the other published studies. One review summarizing this concept is that of Ferrigo et al. (Molecules 2016, 21, 627; doi:10.3390/molecules21050627).

Few examples of such impact of fungal interactions are:

Fumonisin production has been found generally reduced in competing interactions, whereas zearalenone was not affected and DON was increased in co-inocultation experiments (Velluti, A.; Marín, S.; Gonzalez, R.; J Ramos, A.; Sanchis, V. Fumonisin B1, zearalenone and deoxynivalenol production by Fusarium moniliforme, F. proliferatum and F. graminearum in mixed cultures on irradiated maize kernels. J. Sci. Food Agric. 2001, 81, 88–94.) It was demonstrated that high levels of F. verticillioides do not necessarily result in high levels of fumonisin contamination (Ferrigo, D.; Raiola, A.; Causin, R. Plant stress and mycotoxin accumulation in maize. Agrochimica 2014, 58, 116–127.) Insensitivity of ZEA and DON producers to competition occurred when F. graminearum was cultivated with Aspergillus parasiticus, and the toxin levels were not modified (Etcheverry, M. Aflatoxin B1, zearalenone and deoxynivalenol production by Aspergillus parasiticus and Fusarium graminearum in interactive cultures on irradiated corn kernels. Mycopathologia 1998, 142, 37–42)”

A1: We have included a paragraph elaborating on fungal co-occurrence and the effects on fungal growth and mycotoxin production and added 7 more references, including the ones listed by the reviewer: P12, L3-16; Ref. 92-98, P25, L1-18.

Q2: “My two minor comments concern:

Page 5 Line10 : Please complete ‘to cover most Fusarium mycotoxins that were included in the LC-MS/MS analysis’ as following ‘to cover most Fusarium producers of mycotoxins that were included in the LC-MS/MS analysis’”

A2: Adapted in text (P5, L10)

Q3: “Page 11 Line 45 : Replaced the upper-case by a lower-case for ‘And’ in the sentence :’ F. culmorum is correlated with NIV; And the main’”

A3: Adapted in text (P11, L45)